# OFFER PERSONALIZATION USING TEMPORAL CONVOLUTION NETWORK AND OPTIMIZATION

## ABSTRACT

Lately, personalized marketing has become important for retail/e-retail firms due to significant rise in online shopping and market competition. Increase in online shopping and high market competition has led to an increase in promotional expenditure for online retailers, and hence, rolling out optimal offers has become imperative to maintain balance between number of transactions and profit. In this paper, we propose our approach to solve the offer optimization problem at the intersection of consumer, item and time in retail setting. To optimize offer, we first build a generalized non-linear model using Temporal Convolutional Network to predict the item purchase probability at consumer level for the given time period. Secondly, we establish the functional relationship between historical offer values and purchase probabilities obtained from the model, which is then used to estimate offer-elasticity of purchase probability at consumer item granularity. Finally, using estimated elasticities, we optimize offer values using constraint based optimization technique. This paper describes our detailed methodology and presents the results of modelling and optimization across categories.

## 1 INTRODUCTION

In most retail settings, promotions play an important role in boosting the sales and traffic of the organisation. Promotions aim to enhance awareness when introducing new items, clear leftover inventory, bolster customer loyalty, and improve competitiveness. Also, promotions are used on a daily basis in most retail environments including online retailers, supermarkets, fashion retailers, etc. A typical retail firm sells thousands of items in a week and needs to design offer for all items for the given time period. Offer design decisions are of primary importance for most retail firms, as optimal offer roll out can significantly enhance the business' bottom line.

Most retailers still employ a manual process based on intuition and past experience of the category managers to decide the depth and timing of promotions. The category manager has to manually solve the promotion optimization problem at consumer-item granularity, i.e., how to select an optimal offer for each period in a finite horizon so as to maximize the retailer's profit. It is a difficult problem to solve, given that promotion planning process typically involves a large number of decision variables, and needs to ensure that the relevant business constraints or offer rules are satisfied. The high volume of data that is now available to retailers presents an opportunity to develop machine learning based solutions that can help the category managers improve promotion decisions.

In this paper, we propose deep learning with multi-obective optimization based approach to solve promotion optimization problem that can help retailers decide the promotions for multiple items while accounting for many important modelling aspects observed in retail data. The ultimate goal here is to maximize net revenue and consumer retention rate by promoting the right items at the right time using the right offer discounts at consumer-item level. Our contributions in this paper include a) Temporal Convolutional Neural Network architecture with hyperparameter configurations to predict the item purchase probability at consumer level for the given time period. b) Design and implementation of $F_1$-maximization algorithm which optimises for purchase probability cut-off at consumer level. c) Methodology to estimate offer elasticity of purchase probability at consumer item granularity. d) Constraint based optimization technique to estimate optimal offers at consumer-item granularity.

## 2 RELATED WORK

There has been a significant amount of research conducted on offer-based revenue management over the past few decades. Multiple great works have been done to solve Promotion Optimization problem. One such work is Cohen et al. (2017), where the author proposes general classes of demand functions (including multiplicative and additive), which incorporates post-promotion dip effect, and uses Linear integer programming to solve Promotion Optimization problem. In one of the other work [Cohen & Perakis (2018)], the author lays out different types of promotions used in retail, and then formulates the promotion optimization problem for multiple items. In paper Cohen & Perakis (2018), the author shows the application of discrete linearization method for solving promotion optimization. Gathering learnings from the above papers, we create our framework for offer optimization. The distinguishing features of our work in this field include (i) the use of a non-parametric neural network based approach to estimate the item purchase probability at consumer level, (ii) the establishment of the functional relationship between historical offer values and purchase probabilities, and (iii) the creation of a new model and efficient algorithm to set offers by solving a multi-consumer-item promotion optimization that incorporates offer-elasticity of purchase probability at a reference offer value

## 3 METHODOLOGY

We built seperate models for each category, as we understand that consumer purchase pattern and personalized marketing strategies might vary with categories.

### 3.1 MODELLING

In our model setup, we treat each relevant consumer-item as an individual object and shape them into bi-weekly time series data based on historical transactions, where the target value at each time step (2 weeks) takes a binary input, 1/0 (purchased/non purchased). *Relevancy* of the consumer-item is defined by items transacted by consumer during training time window. Our *Positive samples* (purchased/1) are time steps where consumer did transact the item, whereas *Negative samples* (non purchased/0) are the time steps where the consumer did not buy that item. We apply sliding windows testing routine for generating out of time results. The time series is split into 3 parts - train (48 weeks), validation (2 weeks) and test (2 weeks). All our models are built in a multi-object fashion for an individual category, which allows the gradient movement to happen across all consumer-item combinations split in batches. This enables cross-learning to happen across consumers/items. A row in time series is represented by

$$y_{\text{cit}} = h(i_\text{t}, c_\text{t}, .., c_\text{t-n}, ic_\text{t}, .., ic_\text{t-n}, d_\text{t}, .., d_\text{t-n}) \tag{1}$$

where $y_{\text{cit}}$ is purchase prediction for consumer 'c' for item 'i' at time 't'. 'n' is the number of time lags. $i_\text{t}$ denotes attributes of item 'i' like category, department, brand, color, size, etc at time 't'. $c_\text{t}$ denotes attributes of consumer 'c' like age, sex and transactional attributes at time 't'. $c_\text{t-n}$ denotes the transactional attributes of consumer 'c' at a lag of 't-n' time steps. $ic_\text{t}$ denotes transactional attributes such as basket size, price, offer, etc. of consumer 'c' towards item 'i' at time 't' . $d_\text{t}$ is derived from datetime to capture trend and seasonality at time 't'.

#### 3.1.1 FEATURE ENGINEERING

Based on available dataset, we generate multiple features for the modelling activity. Some of the feature groups we perform our experiments are:

**Datetime:** We use transactional metrics at various temporal cuts like week, month, etc. Datetime related features capturing seasonality and trend are also generated. **Consumer-Item Profile:** We use transactional metrics at different granularities like consumer, item, and consumer-item. We also create features like Time since first order, Time since last order, time gap between orders, Reorder rates, Reorder frequency, Streak - user purchased the item in a row, Average position in the cart, Total number of orders. **Price/Promotions:** We use relative price and historical offer discount percentage to purchase propensity at varying price, and discount values. **Lagged Offsets:** We use statistical rolling operations like mean, median, variance, kurtosis and skewness over temporal regressors for different lag periods to generate offsets.

### 3.1.2 LOSS FUNCTION

Since we are solving Binary Classification problem, we believe that Binary Cross-Entropy should be the most appropriate loss function for training the models. We use the below formula to calculate Binary Cross-Entropy:

$$H_{\mathrm{p}} = -\frac{1}{N}\sum_{i=1}^{N} y.log(p(y)) + (1-y).log(1-p(y)) \tag{2}$$

here $H_{\mathrm{p}}$ represents computed loss, y is the target value (label), and p(y) is the predicted probability against the target. The BCELoss takes non-negative values. We can infer from Equation 3 that Lower the BCELoss, better the Accuracy.

### 3.1.3 MODEL ARCHITECTURE

Traditional machine learning models may not be a suitable choice for modelling $h$ (Equation 1) due to non-linear interactions between the features. Sequence to Sequence [Sutskever et al. (2014)] neural network architectures seems to be sound choice for tackling our problem. Hence, we use Entity Embeddings [Guo & Berkhahn (2016)] + Temporal Convolutional Network (TCN) (Figure 1) architecture for building all the models across categories. Originally proposed in [Lea et al. (2016)], TCN can take a sequence of any length and map it to an output sequence of the same length. For this to accomplish, TCN uses a 1D fully-convolutional network (FCN) architecture, where each hidden layer is the same length as the input layer, and zero padding of length (kernel size-1) is added to keep subsequent layers the same length as previous ones. Also, the convolutions in the architecture are causal, meaning that there is no information leakage from future to past. To achieve this, TCN uses causal convolutions [Bai et al. (2018)], convolutions where an output at time t is convolved only with elements from time t and earlier in the previous layer. For 1-D sequence input x and filter f the dilated convolution operation DC on element k of the sequence is defined as:

$$DC(k) = (x *_{\mathrm{d}} f)(k) = \sum_{i=0}^{n-1} f(i) \cdot x_{\mathrm{k-d_i}} \ , where \ x \in \mathcal{R}^n \ and \ f : \{0, ..., n-1\} \to \mathcal{R} \tag{3}$$

where d is the dilation factor, n is the filter size, and $k$-$d_{\mathrm{i}}$ accounts for the direction of the past. When d = 1, a dilated convolution reduces to a regular convolution. Using larger dilation enables an output at the top level to represent a wider range of inputs, thus effectively expanding the receptive field of a ConvNet.

As can be seen in Figure 1, Our network architecture comprises of 3 dilated Convolutions combined with entity embeddings [Guo & Berkhahn (2016)]. Post Convolutions and concatenation with embedding tensor, the created tensor flows through 3 fully connected ReLU layers yielding to sigmoid dense layer. To seggregate static and temporal features, we group input tensor into 4 seperate tensors, as can be seen in 1:

**Static Categorical:** These are categorical features that do not vary with time. This includes consumer attributes like sex, marital status and location along with different item attributes like category, department and brand. **Temporal Categorical:** These are categorical features that vary with time. It includes all the datetime related features like week, month of year, etc. **Static Continuous:** These features are static but continuous. This includes certain consumer attributes like age and weight, item attributes like size, and certain derived features like target encoded features. **Temporal Continuous:** These are time varying continuous features. All consumer and item related traditional attributes like number of orders, add to cart order, etc. falls under this bucket.

### 3.1.4 HYPERPARAMETER TUNING

We use documented best practices along with our experimental results to choose model hyperparameters. Hyperparameter Optimization is performed over validation dataset. We list some of the hyperparameters along with the values we tune for Deep neural network models.

**Optimizer Parameters:** RMSProp [Bengio & CA (2015)] and Adam are used as optimizers across model runs. The learning rate is experimentally tuned to 1e-3. We also have weight decay of 1e-5 which helps a bit in model Regularization. **Scheduler Parameters:** CyclicLR [Smith (2017)] and ReduceLROnPlateau [Zaheer et al. (2018)] Learning rates are used as schedulers across model runs. we use 1e-3 as max lr and 1e-6 as base lr for cyclical learning rate along with the step size being the function of length of train loader. ReduceLROnPlateau is tuned at 1e-6 as min lr. **SWA:** Stochastic

Figure 1: Temporal Convolutional Network (TCN)

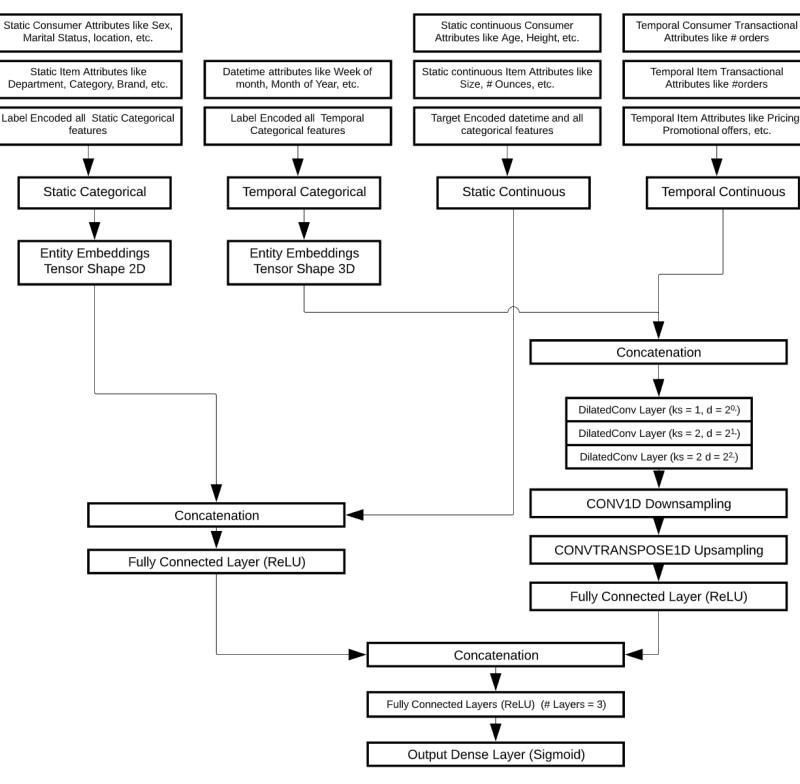

Weight Averaging (SWA) [Izmailov et al. (2018)] is used to improve generalization across Deep Learning models. SWA performs an equal average of the weights traversed by SGD with a modified learning rate schedule. We use 1e-3 as SWA learning rate. **Parameter Average:** This is a method to average the neural network parameters of n best model checkpoints post training, weighted by validation loss of respective checkpoints.

Apart from these parameters we also iterate to tune network parameters like number of epochs, batch size, number of Fully Connected Layers, convnet parameters (kernel size, dilations, padding) and embedding sizes for the categorical features. Binary Cross-Entropy 3 is used as loss function for all the models trained across categories. Neural Network models are built using deep learning framework PyTorch [Paszke et al. (2017)], and are trained on GCP instance containing 6 CPUs and a single GPU.

### 3.2 $F_1$-MAXIMIZATION

Post stacking, we optimize for purchase probability threshold based on probability distribution at a consumer level using $F_1$-Maximization. This enables optimal thresholding of consumer level probabilities to maximize $F_1$ measure [Lipton et al. (2014)]. To illustrate the above, let us say we generated purchase probabilities for 'n' items out of 'b' actually purchased items by consumer 'c'. Now, let us visualize the actuals and predictions (4) of 'n' predicted items for consumer 'c'.

$$A_c = [a_1, a_2, .., a_n] \; \forall \; a_j \in \{0,1\} \;\;, \;\; P_c = [p_1, p_2, .., p_n] \; \forall \; p_j \in \; [0,1] \tag{4}$$

$A_c$ represents the actuals for consumer 'c', with $a_j$ being 1/0 (purchased/non purchased). $P_c$ represents the predictions for consumer 'c' for the respective item, with $p_j$ being probability value. 'n' represents total items for which the model generated purchase probabilities for consumer 'c'. Now we apply Decision rule D() which converts probabilities to binary predictions, as described below in Equation 5.

$$D(Pr_c) : P_c^{\,1 \, x \, n} \rightarrow P_c^{'\,1 \, x \, n} \;\; : p_j^{'} = \begin{cases} 1 & p_j \geq Pr_c \\ 0 & \text{Otherwise} \end{cases} \tag{5}$$

$$P'_c = [p'_1, p'_2, .., p'_n] \ \forall \ p'_j \in \{0,1\} \quad, \quad k = \sum_{i=1}^{n} p'_i \tag{6}$$

$Pr_c$ is the probability cut-off to be optimized for maximizing $F_1$ measure [Lipton et al. (2014)] for consumer 'c'. Decision rule D() converts probabilities $P_c$ to binary predictions $P'_c$ such that if $p_j$ is less than $Pr_c$ then $p'_j$ equals 0, otherwise 1. 'k' is the sum of predictions generated post applying Decision rule D(). Now we solve for $F_1$ measure using equations and formulae described below.

$$V_{Pr_c} = P'_c \ \times \ A_c{}^T \ \Rightarrow \ \begin{pmatrix} p'_1 & .. & p'_n \end{pmatrix} \times \begin{pmatrix} a_1 \\ .. \\ a_n \end{pmatrix} \tag{7}$$

$$Precision_c = \frac{V_{Pr_c}}{k} \quad, \quad Recall_c = \frac{V_{Pr_c}}{b} \quad, \quad F_{1_c} = \frac{2 \times Precision_c \times Recall_c}{Precision_c + Recall_c} \quad \Rightarrow \quad 2 * \frac{V_{Pr_c}}{k+b} \tag{8}$$

$V_{Pr_c}$ represents the number of items with purchase probabilities greater than $Pr_c$ which were actually purchased (True Positives). As can be seen, Formulae 8 is used to calculate Precision, Recall and $F_1$-score for consumer 'c'.

$$\max_{V_{Pr_c}} \quad 2 * \frac{V_{Pr_c}}{k+b} \quad , \quad \text{subject to:} \quad Pr_c \in (0,1) \tag{9}$$

Equation 9 represents the optimization function we solve to generate purchase predictions (1/0) for each consumer. Figure 5 - Section 4 shows the predicted probability distributions.

### 3.3 ELASTICITY FRAMEWORK

After modelling, we establish the functional relationship between historical offer values and purchase probabilities obtained from the model, which is then used to estimate offer-elasticity of purchase probability at consumer item granularity. Given that our output layer of deep net is sigmoid and we are modelling for probability values, sigmoid function (Figure 2) seemed to us as an apt choice to study the variation of purchase probability with offer percent. We also perform multiple experiments as described in Figure 4 - Section 4 to see the goodness of fit of sigmoid curve over our dataset across different categories. The average $R^2$ value for 8 categories is seen to be approximately 75 percent.

$$f(x) = \frac{1}{1 + e^{-(ax+b)}} \quad , \quad f'(x) = a \times f(x) \times (1 - f(x)) \tag{10}$$

Since the functional relationship might vary with categories, we learn seperate parameters of sigmoid for each category. We then use sigmoid curve to estimate elasticities, the x-elasticity of y measures the fractional response of y to a fraction change in x, which can be written as:

$$x - elasticity \ of \ y : \epsilon(x,y) = \frac{dy/y}{dx/x} \tag{11}$$

We incorporate equation 11 to determine the offer-elasticity of purchase probability. We use historical offer percent values at consumer-item granularity, identified using following criteria in that order: a) average of last 4 weeks non-zero offer percent values of the consumer-item combination b) average of historical non-zero offer percent values of the consumer-item combination c) average of last 4 weeks non-zero offer percent values of all same age consumer-item combinations within that category. Using Equations 10 and 11, we establish the offer-elasticity of purchase probability (equation 12) as shown below, k being the offer percent and f(k) being purchase probability.

$$\epsilon(k, f(k)) = f'(k) \times \frac{k}{f(k)} \quad \Rightarrow \quad \epsilon(k, f(k)) = a \times k \times (1 - f(k)) \tag{12}$$

### 3.4 OFFER OPTIMIZATION

Post estimation of offer-elasticity of purchase probability, for each category, we solve the below optimization function (Equation 13) to maximize Net Revenue, with Consumer Retention Rate greater

Figure 2: Purchase probability vs. Offer percentage

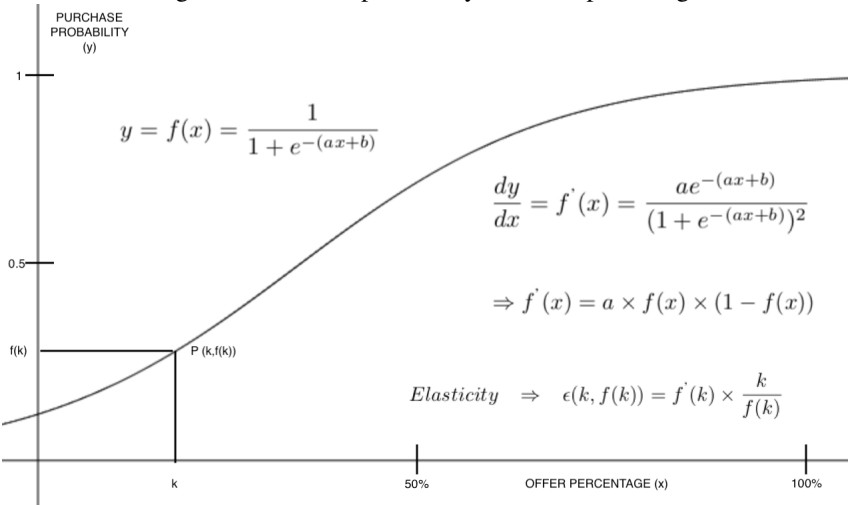

than category threshold ($R_{rc}$).

$$\max_{\eta_i} \quad \sum_{i=1}^{n}[I_p - \frac{I_p}{100} \times (k_i + \eta_i \times k_i)] \times \mathbb{1}_{Pr_c}(f(k_i) + \eta_i \times \epsilon(k_i, f(k_i)) \times f(k_i))$$

$$\text{s.t.} \quad : \quad \eta_i = 0.05 \times j \quad \forall \; j \; \in \; \mathbb{Z} \; and \; j \in (-20, 20)$$

$$(k_i + \eta_i \times k_i) \in (o_1, o_2)$$

$$\frac{1}{n}\sum_{i=1}^{n} \mathbb{1}_{Pr_c}(f(k_i) + \eta_i \times \epsilon(k_i, f(k_i)) \times f(k_i)) \quad >= \quad R_{rc} \tag{13}$$

$$\mathbb{1}_{Pr_c}(x) := \begin{cases} 1 & x \geq Pr_c \\ 0 & \text{Otherwise} \end{cases}$$

In the equation above, $n$ is the total consumer-item samples modelled for a particular category. $I_p$ is the price of the item, $k_i$ and $f(k_i)$ being the offer percent and purchase probability for the $i^{th}$ consumer-item. $\eta_i$ is the change in offer percent $k_i$. $\mathbb{1}_{Pr_c}()$ denotes the Indicator function at $Pr_c$, which is the optimal probability cut-off obtained from $F_1$-Maximization algorithm for consumer 'c' (Equation 9). $\epsilon(k_i, f(k_i))$ denotes the $k_i$ percent offer-elasticity of $f(k_i)$ purchase probability. $R_{rc}$ denaotes the Retention rate cut-off of category $c$. $o_1$ and $o_2$ refers to offer range for that category. This is determined using the latest 2 weeks of consumer-item samples. We solve the Equation 13 using Linear Programming approach to compute optimal offer at consumer-item granularity. We observe that there is variance in the optimal offers generated from optimization engine across categories.We have shown the distribution of offers across categories in Figure 6 - Section 4.

## 4 EXPERIMENTS AND RESULTS

We use kaggle dataset available at AmExpert 2019. Figure 7 shows the schema of the data we have used to build our models. As mentioned in Section 3, we use a maximum of 1 year data aggregated at bi-weekly level at consumer-item granularity. We split the data into three part train, validation and test based on our validation strategy. We generate consumer - item - bi-weekly data with purchase/ non purchase being the target , and use this data to train all our models.

### 4.1 EXPERIMENT SETUPS

We start with exploratary data analysis, looking at the data from various cuts. We study the variations of different features with our target (purchase/ non purchase). Some of our studies include category-wise sales distribution, density of transactions with varying offer percent and density of

Figure 3: Exploratory Data Analysis

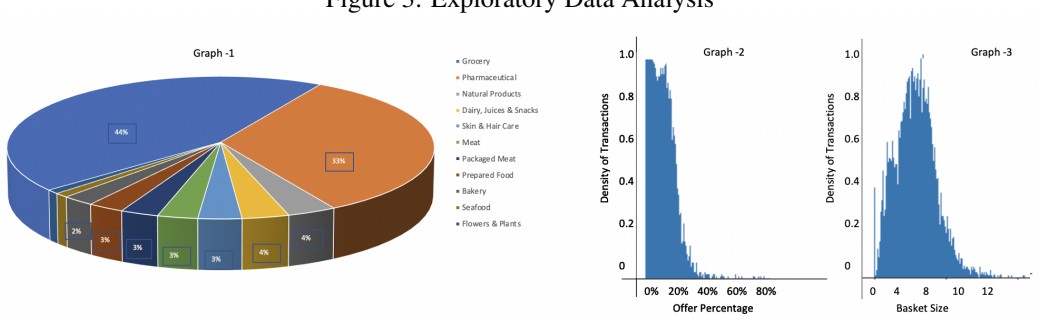

**Graph-1: Category-wise sales distribution, Graph-2: density of transactions with varying offer percent, and Graph-3: density of transactions with varying basket size**

transactions with varying basket size (Figure 3). We observe that Grocery and Pharmaceutical contributes to approximately 77 percent of total sales. We also look at order probability variations at different temporal cuts like week, month and quarter, transactional metrics like total orders, total reorders, recency, gap between orders, at both consumer and item levels. We then perform multiple experiments with the above mentioned features and different hyperparameter configurations to land at reasonable hyperparameters to perform final experiments and present our results.

Figure 4: Offer-elasticity of purchase probability

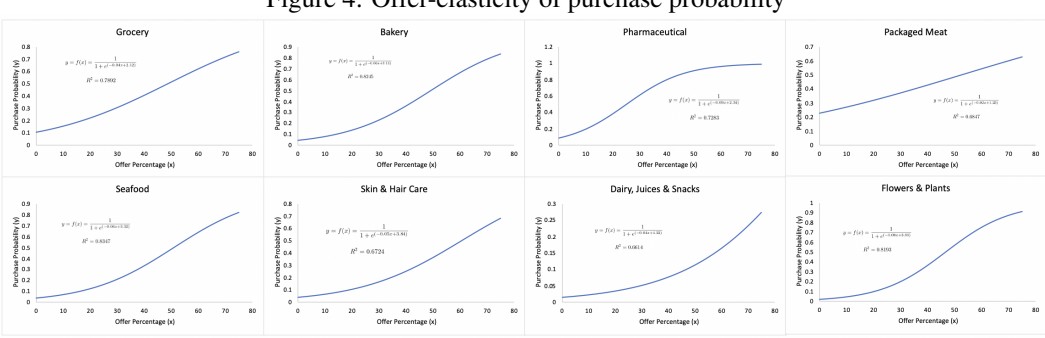

**Category level visualizations describing the sensitivity of change in offer with purchase probability**

Figure 5: Probability Distributions

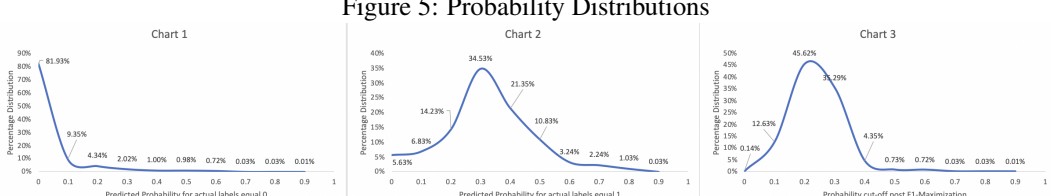

**Chart-1: Predicted probability distribution at actual label equals 0, Chart-2: Predicted probability distribution at actual label equals 1, and Chart-3: cut-off probability distribution post $F_1$-Maximization**

## 4.2 RESULTS AND OBSERVATIONS

Figure 4 shows the functional relationship of offer percetage with purchase probability obtained across different categories. It also contains the equations and goodness of fit in the form of $R^2$ for each category. Seafood and Bakery are the top 2 best fitted categories with $R^2$ values of 0.83 and 0.82 respectively. Figure 5 Chart-1 and Chart-2 shows the predicted probability distribution when actual label equals 0 and 1 respectively over the validation data split. Chart-3 shows the cut-off probability

distribution post $F_1$-Maximization, and, we observe that highest density of cut-off probability lies between 0.2 to 0.3. Table 1 presents the accuracy values post Modelling and $F_1$-Maximization. It also has the average elasticity values along with weighted offer percent computed post optimization. From model performance perspective , it is observed that Grocery category has least BCELoss of 0.0283. Pharmaceutical and Meat categories followed Grocery with BCELoss of 0.0296 and 0.0299 respectively. Also, Grocery has the best $F_1$ score of 0.512 followed by Meat and Pharmaceutical scoring 0.511 and 0.509 respectively. Vegetables is the most elastic category with elasticity value of 1.53, whereas Packaged meat and Skin and Hair care are the least elastic categories with elasticity value of 0.62. From Figure 6, we can see the graphical representation of distribution of optimal offer calculated through optimization. We find Skin and Hair care along with Pharmaceutical to be left skewed, whereas Vegetables as well as Flowers and Plants to be right skewed.

Figure 6: Consumer-Item percent distribution with offer percent across categories

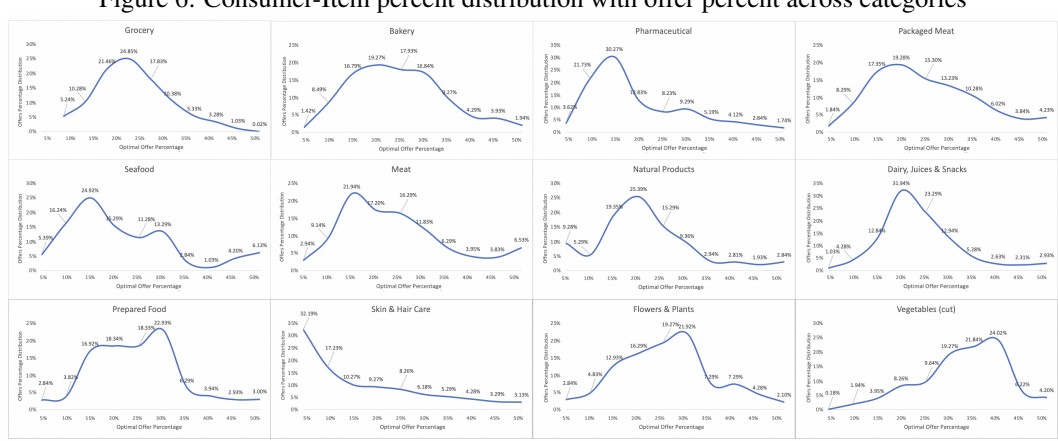

**Category level visualizations of frequency distribution of offers generated by optimization engine**

Table 1: Model Results and Elasticity

| Category | Sample size | BCELoss | Precision | Recall | $F_1$-Score | Avg Elasticity | Weighted Offer Percent |
|---|---|---|---|---|---|---|---|
| Grocery | 28,990 | **0.0283** | **0.524** | **0.501** | **0.512** | **1.13** | 23.37 |
| Bakery | 1,628 | 0.0415 | 0.346 | 0.391 | 0.367 | 1.06 | 22.15 |
| Pharmaceutical | 20,492 | **0.0296** | **0.538** | 0.483 | **0.509** | 0.73 | 17.89 |
| Packaged Meat | 1,473 | 0.0329 | 0.473 | 0.452 | 0.462 | 0.62 | 19.15 |
| Seafood | 539 | 0.0382 | 0.497 | 0.379 | 0.43 | 0.89 | 16.23 |
| Natural Products | 2,399 | 0.0319 | 0.451 | **0.524** | 0.485 | 0.95 | 21.78 |
| Dairy, Juices, Snacks | 2,060 | 0.0396 | 0.394 | 0.492 | 0.438 | 1.04 | 22.02 |
| Prepared Food | 1,407 | 0.0472 | 0.338 | 0.295 | 0.315 | 1.03 | **28.27** |
| Skin, Hair Care | 1,906 | 0.0391 | 0.429 | 0.453 | 0.441 | 0.62 | 7.93 |
| Meat | 1,767 | **0.0299** | 0.498 | **0.524** | **0.511** | 0.94 | 18.92 |
| Flowers, Plants | 491 | 0.0483 | 0.275 | 0.318 | 0.295 | **1.43** | **30.24** |
| Vegetables (cut) | 124 | 0.0469 | 0.364 | 0.267 | 0.308 | **1.53** | **37.92** |

## 5 CONCLUSION

We have presented our detailed methodology to solve the offer optimization problem at the intersection of consumer, item and time in retail setting. We have also presented the results of our models and optimization in the form of model accuracies and graphical representations at a category level. At the same time we understand that computation strategy is a key aspect in modelling millions of consumers, and we intend to further explore this aspect by building Transfer Learning framework Yosinski et al. (2014). We are also working to further improve our Sequence to Sequence neural network architectures to improve accuracy and decrease computation time.

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

# A  APPENDIX

Figure 7: Data Schema

