# OpenReview forum: "OFFER PERSONALIZATION USING TEMPORAL CONVOLUTION NETWORK AND OPTIMIZATION"
_ICLR.cc/2021/Conference — Reject_

### Official Review · AnonReviewer2 · 2020-10-27
**integrated work and neat model, but have some concerns**

**Rating:** 4
**Confidence:** 2

**Review:**

This paper uses TCN structure and try to understand consumer purchase pattern for personalized marketing strategies. The corresponding optimization optimizes the purchase probability. However, there are several concerns.

1, The motivation is not clear. The contribution of the proposed model doesn't support personalization capacity. There is no definition of personalization in the promotion settings. I didn't understand how you involve the personalization in either eq1 or section 3.3-2.4.

2, The problem formulation is not clear, the goal of the designed model is targeted at max Revenue or supporting the personalization or max F1? If both, how to evaluate and compare that the proposed method is able to provide more personalized results?

3, Some conceptions and statements are not clear, what is consumer purchase pattern? what is personalized market strategies? how to formulate these conceptions?

4, why you choose BCE loss function instead of Hinge loss function?

5, It seems the sensitivity of change in offer with purchase probability you provided in figure 4 has take a dominate role in maximizing the revenue. In order to consider the personalization, does it associate with different consumers because of potential different purchase patterns even within same category?

---

### Official Review · AnonReviewer1 · 2020-10-28
**Interesting application but limited technical contribution**

**Rating:** 3
**Confidence:** 4

**Review:**

##########################################################################

Summary:
- This work proposed a framework to optimize and personalize deal-based promotion strategy. The proposed framework is composed of a Temporal Convolutional Network to predict purchase probability, a function to approximate price elasticity, a constraint based utility function to optimize promotion strategy. The proposed method is tested on a public dataset.

##########################################################################

Strength:
- This work is trying to address a real-world problem and has posed a reasonable framing
- Indeed all three components of this framework are important research areas and the overall problem is also a critical application scenario
- The overall presentation is easy to follow

##########################################################################

Weakness:
- Limited technical novelty of the proposed methods
- The overall execution needs significant improvements
- A lot of related work is missing

##########################################################################

Detailed Comments:

Despite the importance of the problem, unfortunately the overall technical novelty and execution of this work does not reach the standard of an ICLR research paper - therefore I vote for a clear *rejection*.

- From the methodology perspective, as mentioned in the previous comments, all three components in this problem framing are important research areas - while the proposed framework in this paper is more like a combination of three existing methods. In this regard, I didn't find significant technical contributions out of this.

- Following this, a lot of previous work in the abovementioned three areas, especially the first two areas need to be acknowledged in the related work. To name a few (some work I'm aware of but not limited to)
 - a) Modeling purchase probability with price sensitivity
 [1] "Modeling Consumer Preferences and Price Sensitivities from Large-Scale Grocery Shopping Transaction Logs", WWW'17
 [2] "Shopper: A probabilistic model of consumer choice with substitutes and complements", Annals of Applied Statistics (2017)
 [3] "Price-aware Recommendation with Graph Convolutional Networks", ICDE'20
and potentially many papers in the recommender system area leveraging the Temporal Convolutional Network
 - b) Causal inference for price elasitisty (unbiased characterization between demand and price)
 [4] "Estimation and Inference about Heterogeneous Treatment Effects in High-Dimensional Dynamic Panels" Semenova, Goldman, Chernozhukov, Taddy (2017)
Additional work from economics and marketing community can also be considered.

- Unfortunately the execution of this work needs many further improvements. The most critical problem is the lack of rigorous, quantitative evaluation of the proposed method. Current experiment execution is more like a data analysis on a public dataset - without evaluating against baselines (for example, some of the previous mentioned papers), it is very difficult to justify the effectiveness of the proposed method.

Overall I'd suggest the authors carefully conduct literature reviews in this domain, improve the execution and evaluation, and possibly consider an application-driven venue (e.g. an ecommerce workshop etc.) for this work.

##########################################################################

Typos:
- Period is missing at the end of Related work
- Left quotation marks are not correctly rendered (e.g. in Section 3.1, consider using ` in latex instead of ')

---

### Official Review · AnonReviewer3 · 2020-10-28
**Applications-focused paper on optimizing customer offers,  but lacks baselines.**

**Rating:** 5
**Confidence:** 4

**Review:**

This paper is about optimizing discount offers to individual customers to maximize business value. A temporal convolutional network (TCN) is used to model the customer's purchase probability. This network is then used to estimate the customer's offer-elasticity. Finally a linear program is used to optimize offers across all customers subject to a retention-rate constraint. The method is demonstrated on a public dataset and a detailed analysis is performed on its results.

Strengths of paper:
+ The application of the paper, to optimize personalized offers, has an important real-world business impact.
+ To my knowledge, their framework to use customer modeling to identify offer elasticity is novel.
+ The methods are described in much detail which will help in replication.

Weaknesses of paper:
- The approach is not compared against baseline methods to demonstrate superior performance.
- The paper has limited novelty as the authors use known techniques such as TCN and linear programming.
- The writing can be improved. More detail in comments.

Overall this technique seems to be a good example of how to solve a real-world problem with standard machine learning and optimization tools. However, it needs to be compared against other baselines to really demonstrate value. It would also help to put this technique in greater context to prior work. The "related works" section is quite brief and only cites three other works.

Comments:
1. Notation is overloaded so that the symbol may have different meanings in different sections of the paper. For example, d is both a dilation factor and a date-time feature.
2. It would also be more standard to signify variables by using an appropriate font rather than quotes. ($d$ vs 'd')
3. On page 3, it says "as can be seen in 1:". Is this a broken latex reference?
4. Some mis-spelled words -- "seperate", "denaotes", "percetage"
5. I am unfamiliar with retention rate cutoffs. This could use more explanation.
6. In figure 4, it would be better to use the same y-axis across graphs for easier comparison. As is, they all look to have similar offer elasticity.
7. Why did you choose to optimize for F-1 rather than some other metric?
8. This is a nitpick, but some of your features are "numeric" rather than "continuous" since they can only take on integer values.

---

### Decision · Program_Chairs · 2021-01-07
**Final Decision**

**Decision:**

Reject

**Comment:**

Reviews are somewhat mixed, but all are below the acceptance threshold. Reviewers praise the overall application and the presentation (though there is some variance in response to this aspect), but have concerns about lack of certain comparisons and technical novelty.